# Functional NHE1 expression is critical to blood brain barrier integrity and sumatriptan blood to brain uptake

**Erika Liktor-Busa, Kiera T. Blawn, Kathryn L. Kellohen, Beth M. Wiese, Vani Verkhovsky, Jared Wahl, Anjali Vivek, Seph M. Palomino[ID], Thomas P. Davis, Todd W. Vanderah, Tally M. Largent-Milnes[ID]***

Department of Pharmacology, University of Arizona, Tucson, Arizona, United States of America

* tlargent@email.arizona.edu

**Data Availability Statement:** All relevant data are within the manuscript and its Supporting Information files.

## Abstract

Disruption of blood-brain barrier integrity and dramatic failure of brain ion homeostasis including fluctuations of pH occurs during cortical spreading depression (CSD) events associated with several neurological disorders, including migraine with aura, traumatic brain injury and stroke. NHE1 is the primary regulator of pH in the central nervous system. The goal of the current study was to investigate the role of sodium-hydrogen exchanger type 1 (NHE1) in blood brain barrier (BBB) integrity during CSD events and the contributions of this antiporter on xenobiotic uptake. Using immortalized cell lines, pharmacologic inhibition and genetic knockdown of NHE1 mitigated the paracellular uptake of radiolabeled sucrose implicating functional NHE1 in BBB maintenance. In contrast, loss of functional NHE1 in endothelial cells facilitated uptake of the anti-migraine therapeutic, sumatriptan. In female rats, cortical KCl but not aCSF selectively reduced total expression of NHE1 in cortex and PAG but increased expression in trigeminal ganglia; no changes were seen in trigeminal nucleus caudalis. Thus, *in vitro* observations may have a significance *in vivo* to increase brain sumatriptan levels. Pharmacological inhibition of NHE1 prior to cortical manipulations enhanced the efficacy of sumatriptan at early time-points but induced facial sensitivity alone. Overall, our results suggest that dysregulation of NHE1 contributes to breaches in BBB integrity, drug penetrance, and the behavioral sensitivity to the antimigraine agent, sumatriptan.

## Introduction

The Headache Classification Committee of the International Headache Society defined migraine as a recurrent headache characterized by unilateral location, pulsating quality, and moderate to severe intensity, which is accompanied with nausea and/or photo-phonophobia [1,2]. In approximately one third of migraine patients, episodes are associated with unilateral, fully reversible, visual, sensory or other CNS symptoms that usually develop gradually [3]; this symptomology is collectively termed migraine aura [4]. Several clinical and neuroimaging findings support a pathophysiological connection between cortical spreading depression

**Funding:** This work was supported by grants from the National Institute of Neurological Disorders and Stroke (R01NS099292, TML) of the National Institutes of Health, Arizona Biomedical Research Commission (ABRC45952, TML), and with monies from the Department of Pharmacology at the University of Arizona. Authors are solely responsible for the content which does not necessarily represent the official views of the National Institutes of Health, the State of Arizona, or the University of Arizona. The funders had no role in study design, data collection and analysis, decision to publish, or preparation of the manuscript.

**Competing interests:** The authors have declared that no competing interests exist.

(CSD) and migraine aura [5–7]. In addition to migraine with aura, CSD can be induced by traumatic brain injury, hemorrhage or ischemia, and can develop over the course of epileptic seizure [8]. CSD is an intense self-propagating depolarization wave originated from cerebral gray matter that propagates slowly across the brain (2–5 mm/min) [9] that is followed by a longer lasting wave of hyperpolarization characterized by massive flux in ionic concentrations and limited neurotransmitter release [10]. CSD events can cause a triphasic perturbation of extracellular pH, manifested as a small initial acidic shift, followed by a rapid transient alkaline shift then a large and prolonged tissue acidosis which is coupled to neuronal but not astrocytic swelling and decreases in intracellular pH [11–14]. MRI study on a case of a familial hemiplegic migraine patient demonstrated a drop in pH during prolonged aura, further supporting the clinical relevance of pH changes during CSD events [15].

The blood-brain barrier (BBB) is a dynamic and functional interface that separates the central nervous system from peripheral circulation. Several publications suggest that BBB plays a crucial role in maintaining proper neuronal function. Disruption of BBB integrity has been reported after direct and indirect insult, including peripheral inflammatory and neuropathic pain conditions [16–18]. Rodent models of spreading depression revealed evidence that CSD can initiate a cascade that increases BBB permeability [19,20]. Fried et al. findings suggest a region-specific enhancement of BBB permeability in brain areas involved in trigeminal pain during episodic and chronic stages of repeated, inflammatory, dural stimulation, supporting a disrupted BBB in migraine with aura [21].

*In vitro*, shifts in pH are linked to opening of the BBB. A primary gene family maintaining intracellular pH (pHi) is *SLC9*. Three members of *SLC9*/NHE (sodium-hydrogen exchanger) family, namely NHE1, NHE3, and NHE5, coded by *SLC9A1*, *SLC9A3*, and *SLC9A5* genes, respectively, mediate electroneutral change of one $Na^+$ for one $H^+$ across plasma membranes [22]. Among those, NHE1 plays a crucial role in the regulation of intracellular pH in neurons and endothelial cells [23–25]. Previous studies have shown that pharmacological inhibition of NHE1 protein with its inhibitors has neuroprotective effects in experimental stroke models and prevents BBB damage [26–29]. To date, changes in NHE1 expression and function after CSD induction have not been studied.

It is known that the pH is a regulator of 1) BBB integrity [30–32] and 2) transport of xenobiotics across the BBB [33,34]. Mounting evidence suggest that CSD induces regional perturbations in pH [35], but the direct connection between pH change induced by CSD event and disrupted BBB integrity has not been defined yet. The overall goals of the studies herein focused on NHE1 as one of the main regulators of pH and how it is implicated in cortical KCl-induced deficiencies of BBB integrity resulting from pH dysfunction and regulation of sumatriptan blood to brain uptake. Results indicate that BBB paracellular integrity requires functional NHE1 expression. Moreover, sumatriptan uptake *in vitro* and analgesic effects *in vivo* were enhanced when NHE1 function was impaired. Together, these data indicate a vital role that NHE1 expression and function plays at the BBB and highlights that comprise of NHE1 may affect trans-and paracellular routes of BBB in different ways.

## Materials and methods

### Drugs and reagents

Ketamine/xylazine was purchased from Sigma-Aldrich (St. Louis, MO) and isoflurane from VetOne (IL, USA). Zoniporide (SML0076) was purchased from Sigma-Aldrich (St. Louis, MO), dissolved in water at 1 mM concentration for *in vitro* experiments, which was further diluted in the appropriate media. The final concentration of zoniporide was 10 nM. Sumatriptan succinate (S1198) was purchased from Sigma-Aldrich (St. Louis, MO). For *in vivo*

experiments, saline was used as a diluent for zoniporide and sumatriptan with an injection volume of 1 mL/kg. All other chemicals, unless noted were purchased from Sigma-Aldrich (St. Louis, MO).

## Animals

Female Sprague Dawley rats (200–250 g), purchased from Envigo (Indianapolis, IN), were housed in a climate-controlled room on a regular 12 h light/dark cycle with lights on at 7:00 am with food and water available *ad libitum*. Animals were initially housed 3 per cage but were individually housed after dural cannulation. All procedures were performed during the 12-hour light cycle according to the policies/recommendations of the International Association for the Study of Pain and the NIH guidelines for laboratory animals, and with IACUC approval from the University of Arizona.

## Dural cannulation

Dural cannulation was performed as previously described [36,37]. Briefly, anesthesia was induced with intraperitoneal 80:10 mg/kg ketamine:xylazine or 45:5:2 mg/kg cocktail of ketamine:xylazine:acepromazine. Rats were placed in a stereotactic frame (Stoelting Co.), and a 1.5- to 2-cm incision was made to expose the skull. A 0.66- to 1-mm hole (Pinprick/KCl: -6.5 mm A/P, -3 mm M/L from bregma) was made with a hand drill (DH-0 Pin Vise; Plastics One) to carefully expose, but not damage, the dura. A guide cannula (D/V: -0.5 mm from top of skull, 22 GA; Plastics One, C313G) was inserted into the hole and sealed into place with glue. Two additional 1 mm holes were made rostral to the cannula to receive stainless-steel screws (Small Parts, MPX-080-3F-1M), and dental acrylic was used to fix the cannula to the screws. A dummy cannula (Plastics One, C313DC) was inserted to ensure patency of the guide cannula. Rats were housed individually and allowed 6–8 d to recover. Cannula placement and dural integrity at screw placement was confirmed postmortem.

## Cortical injections, treatment with zoniporide and sumatriptan

Cortical injections were performed using a Hamilton injector (30 GA, #80308 701 SN, Hamilton Company) customized to project 1.0 mm into and beyond the dura into the occipital cortex. The injector was inserted through the guide cannula to deliver a focal injection of 0.5 μl of 1 M KCl (~60mM final concentration at site) or artificial CSF (aCSF: 145 mM NaCl, 2.7 mM KCl, 1 mM $MgCl_2$, 1.2 mM $CaCl_2$, and 2 mM $Na_2HPO_4$, pH 7.4), and the solution was passed through a 0.2-μm syringe filter before injection. A second set of animals received dural application of aCSF or KCL (5μL) rather than a cortical injection. Zoniporide was dissolved in saline (1 mL/kg) and injected intraperitoneally at 1 mg/kg or 5mg/kg doses, 10 min before cortical KCl injection. Sumatriptan succinate (0.6 mg/kg, dissolved in saline) was dosed subcutaneously 30 min post-cortical KCl administration. Dural injections of saline or lidocaine (2%) were performed using a customized injector which was flush with the end of the dural guide cannula at a volume of 5μL.

## Periorbital mechanical allodynia

Periorbital allodynia was evaluated before and at 30, 60, 90, 120, and 180 minutes after cortical injection of KCl or aCSF by an observer blinded to drug and cortical conditions. Rats were grouped based on their postsurgical baseline to ensure equivalent pre-injection thresholds (6-8g). Any rats exhibiting excessive postsurgical, but pre-pharmacological manipulation allodynia, (threshold <6 g) were removed from the study. Rats were acclimated to testing box 1 hour

prior to evaluation of periorbital mechanical allodynia with von Frey filaments as previously described [38,39]. Behavioral responses were determined by applying calibrated von Frey filaments perpendicularly to the midline of the forehead at the level of the eyes with enough force to cause the filament to slightly bend while held for 5 seconds. A response was indicated by a sharp withdrawal of the head, vocalization, or severe batting at the filament with attempts to eat it.

## Assessment of fluorescein isothiocyanate (FITC)-dextran extravasation

Fluorescein isothiocyanate (FITC) labeled dextran (4 kDa, Sigma, #46944) was employed as a marker molecule to assess BBB permeability after cortical KCl or aCSF injection in combination with zoniporide. Groups of three-four female rats were randomly allocated into four groups: group 1: aCSF+zoniporide, group 2: KCl+zoniporide, group 3: aCSF+saline, and group 4: KCl+saline. Zoniporide was injected intraperitoneally at 1 mg/kg dose, 10 min before cortical KCl or aCSF injection. In control animals, equivalent volume of saline was injected as vehicle-control. FITC-dextran (5% in saline) was intravenously injected at 10 min before the tissue harvest. The tissue (cortex and serum) was collected at 90 min after cortical injections, following the transcardial perfusion of ice-cold 0.1 M phosphate buffer. The samples were snap-frozen and stored at -80˚C until processing. On the day of processing, the tissue weight was measured, and the cortex samples were sonicated in 0.1 M phosphate buffer. The fluorescence intensity was measured in microplate reader (CLARIOstar Plus) along with a calibration curve. The whole experiment was carried out under low light conditions to minimize quenching of fluorescence of the FITC-dextran solution.

## Tissue collection for Western immunoblotting

Rats were anesthetized with ketamine/xylazine mix as above, then transcardially perfused with ice cold 0.1 M phosphate buffer at rates to not burst microvasculature (i.e., 3.1 mL/min). After decapitation, tissue samples, cortex (Ct), trigeminal ganglia (TG), trigeminal nucleus caudalis (Vc) and periaqueductal grey (PAG) were harvested, flash frozen in liquid nitrogen and stored at -80˚C until preparation. On the day of preparation, samples were placed in ice-cold lysis buffer (20 mm Tris-HCl, 50 mM NaCl, 2 mM MgCl$_2$x6H$_2$O, 1% v/v NP40, 0.5% v/v sodium deoxycholate, 0.1% v/v SDS; pH 7.4) supplemented with protease and phosphatase inhibitor cocktail (BiMake, B14002, and B15002). All subsequent steps were performed on ice or at 4˚C. The samples were sonicated, then centrifuged at 15,000xg for 10 minutes. The supernatant was collected from the samples and BCA Assay was performed to determine the protein content (Pierce™ BCA Protein Assay Kit, Thermo Scientific, 23223).

## Western immunoblotting

Total protein (10 μg) from the tissue supernatant was loaded on TGX precast gels (10% Criterion™, BioRad) and then transferred to nitrocellulose membranes (Amersham™ Protran™, GE Healthcare). After transfer, the membranes were blocked at room temperature for 1 h in blocking buffer (5% BSA in Tris-buffered saline with Tween 20 (TBST)). The following primary antibodies were diluted in blocking buffer: NHE1 (Abcam, ab67313, 1:500), and α-tubulin (Cell Signaling, 3873S, 1:20,000). The membranes were incubated in the diluted primary antibodies overnight at 4˚C. The blots were then washed three times in TBST, incubated with secondary antibodies: GαM680 (LiCor, 926–68020), and GαR800 (LiCor, 926–32211) in 5% milk in TBST for 1 h of rocking at room temperature, washed again, and imaged with a LiCor Odyssey infrared imaging system (LiCor, Lincoln, NE). Un-Scan-It gel version 6.1 scanning software (Silk Scientific Inc.) was used for quantification. To visualize multiple bands on the

same blot, blots were stripped with One Minute® Plus Western Blot Stripping Buffer (GM Biosciences, GM6510). Protein expressions were corrected for the expression of the loading control (e.g. α-tubulin).

## Culture and treatment of immortalized cell lines

bEnd.3 (CRL-2299, ATCC) cells were cultured in DMEM (Gibco, 11995–065), supplemented with 2 mM L-glutamine (ThermoScientific, 25030081), 10% fetal bovine serum (Gibco, 10082139), and penicillin (100 UI/mL)-streptomycin (100 μg/mL) (Invitrogen, 15140122). 24 hours before KCl pulse, the media of bEnd.3 cells were changed to astrocyte-conditioned media harvested from confluent C8-D1A flasks. C8-D1A (CRL-2541, ATCC) and C8-B4 (CRL-2540, ATCC) cells were maintained in DMEM (Gibco, 11995–065), supplemented with 10% fetal bovine serum (Gibco, 10082139), and penicillin (100 UI/mL)-streptomycin (100 μg/mL) (Invitrogen, 15140122). All cell lines were cultivated at 37˚C in a humidified 5% $CO_2$/95% air atmosphere. All cell lines were treated with 60 mM KCl for 5 minutes, which is a typical condition to evoke potassium-triggered spreading depolarization in live brain slices [40]. Samples for subsequent analysis (subcellular fractionation, whole cell lysate, and immunocyto-chemistry) were harvested at 5 and 30 minutes after KCl pulse or aCSF application.

## Primary culture of trigeminal ganglion

Primary culture of trigeminal ganglion was prepared from female rats using a modified proto-col published by Malin et al. [41]. Female Sprague-Dawley rats (4 weeks old) were anesthetized with ketamine and perfused transcardially with ice-cold PBS (pH 7.4). The trigeminal ganglia were dissected and placed in sterile HBSS without $Ca^{++}$/$Mg^{++}$ (ThermoScientific, 14170120). The ganglia were chopped into 10–12 pieces with spring scissors and transferred to a tube containing 1.5 mL HBSS without $Ca^{++}$/$Mg^{++}$. Enzymatic dissociation was performed with 20 U papain/mL (Worthington, 3126) for 20 min at 37˚C, followed by a second enzymatic digestion in 4 mg/ml collagenase type II (Worthington, 4176) and 4.5 mg/ml dispase type II (Worthington, 165859,) for 20 min at 37˚C. The dissociated cells were centrifuged at 400×g for 4 min. The cell pellet was resuspended in warm F12 media (Gibco, 11765) supplemented with 10% FCS (10082139, Invitrogen), and penicillin (100 UI/mL)-streptomycin (100 μg/mL) (Invitrogen, 15140122). The tissue was dissociated by pipetting up and down in a P1000 pipette tip increasingly smaller gauges, with the smallest diameter being about twice that as a normal P1000 tip opening, and the largest diameter being about half that of the widest part of the pipette tip. The final trituration was performed until most of the tissue chunks had been disso-ciated. The cell suspension was passed through a pre-wet 70 μm cell strainer. The cells were plated at a density of 60,000 cells per coverslip coated with poly-D-lysine (20 μg/ml). Then the cells were incubated for 2 h at 37˚C to allow cell adhesion with the volume completed using B-27™ Plus Neuronal Culture System (Gibco, A3653401), supplemented with penicillin (100 UI/mL)-streptomycin (100 μg/mL) (15140122, Invitrogen). The TG cultures were incubated for 5 days, changing culture media every second day. The primary culture of TG was treated with an aCSF or KCl pulse, as in immortalized cells.

## Immunocytochemistry

The cells (bEnd.3, C8-D1A, and C8-B4) cultured on collagenated glass coverslips (10,000 cells/coverslip) were washed twice with PBS (pH 7.2), and fixed in PBS containing 1% PFA, for 10 minutes at room temperature. After fixation, cells were washed in PBS and permeabilized with 0.2% Triton-X-100 in PBS for 10 minutes at room temperature. The cells were then incubated with 10% BSA in 0.1% Triton-X-100 in PBS for 1 hour at room temperature to reduce

nonspecific binding. After 3 washes in PBS, primary antibody against NHE1 (Abcam, ab67313, 1:200 in blocking buffer) were applied overnight at 4°C. Followed by 5 washes in PBS, AlexaFlour™488 secondary antibody (ThermoScientific, 1:500 in blocking buffer) was used for 1 hour at room temperature. After several washes in PBS, the coverslips were mounted to slides with Prolong Gold Antifade with DAPI (ThermoScientific, P36941) mounting media and allowed to dry. The slides were visualized using Olympus microscope (BX61), and images digitally captured (CoolSNAP HQ; Olympus).

## Cell lysis and subcellular fractionation

Membrane, cytoplasmic, and nuclear fractions of bEnd.3 cells treated with KCl pulse (60mM, 5 min) or aCSF (equivolume) were prepared using Subcellular Protein Fractionation Kit for Cultured cells (ThermoScientific, 87790) as per manufacturer's instruction. Each extract (10 µg) was analyzed by Western blot (described above), using NHE1 (Abcam, ab67313, 1:500), α-tubulin (Cell Signaling, 3873S, 1:20,000), lamin B (Invitrogen, PIPA519468, 1:1000), and PECAM (NB100-2284, Novus Biologicals, 1:1000) primary antibodies. The intensity levels of NHE1 in each fraction were normalized to the housekeeping proteins: α-tubulin for the cytosol, lamin B for the nuclear and PECAM for the membrane fractions. The whole cell lysates were prepared in ice-cold lysis buffer (20 mm Tris-HCl, 50 mM NaCl, 2 mM $MgCl_2x6H_2O$, 1% v/v NP40, 0.5% v/v sodium deoxycholate, 0.1% v/v SDS; pH 7.4) supplemented with protease and phosphatase inhibitor cocktails (BiMake), and centrifuged at 13,000xg for 10 minutes at 4°C.

## Isolation of cortical microvessels

Rat cortical microvessels were isolated as previously described [42,43]. Briefly, rats were anesthetized with intraperitoneal injection of 80:10 mg/kg ketamine:xylazine, decapitated and the brains placed in ice-cold buffer A (136.9 mM NaCl; 2.7 mM KCl; 1 mM $CaCl_2$; 1.5 mM $KH_2PO_4$, 8.1 mM $Na_2HPO_4$; 0.5 mM $MgCl_2$; 5 mM glucose; 1 mM sodium pyruvate, pH7.4) supplemented with Roche EDTA-free Complete Protease Inhibitor cocktail, Sigma protease inhibitor cocktail and 2 mM phenylmethylsulfonyl fluoride. All subsequent steps were performed on ice or at 4°C. After removal of the choroid plexis and meninges, three rat brains were pooled and homogenized in 20 ml of buffer A using a Potter-Elvehjem homogenizer with 20 strokes at moderate speed followed by 8 strokes in a glass Dounce homogenizer by hand. Homogenate was mixed with 30% Ficoll in buffer A and centrifuged for 20 min at 5800 × g in a Sorvall SS-34 rotor. The pellet was resuspended in 10 ml buffer A supplemented with 1% BSA using 2 strokes with the Potter-Elvehjem homogenizer. The suspension was then filtered through a 300 µm mesh filter and microvessels collected on a 40 µm mesh filter. Microvessels retained on the 40 µm mesh filter were resuspended in buffer A with 1% BSA and pelleted by centrifugation (10 min at 1500 × g). The pellet was washed twice in buffer B (20 mM Tris-HCl; 250 mM sucrose; 1 mM $CaCl_2$; 1 mM $MgCl_2$, pH 7.8) supplemented with Roche EDTA-free Complete Protease Inhibitor cocktail, Sigma protease inhibitor cocktail and 2 mM phenylmethylsulfonyl fluoride, and collected by centrifugation for 10 min at 2170 × g. The microvessel pellet was stored at −80°C until further use.

## Quantification of intracellular and extracellular pH

bEnd.3 cell monolayers were grown on collagen-coated glass bottom culture dish (Electron Microscopy Sciences). On the day of experiment, the cells were washed with Live Cell Imaging Solution (LCIS, Invitrogen) three times and then incubated with 5 µM of BCECF-AM (Invitrogen, B1170) at 37°C for 30 minutes. After several washes with LCIS, the cells were treated with

aCSF for 5 minutes, followed by KCl pulse (60mM) for 5 minutes. Images were taken by Zeiss LSM880 inverted confocal microscope every 10 seconds during the whole treatment period for up to 45 min. Intracellular pH calibration buffer kit (Invitrogen, P35379) was used as per manufacturer's instruction to obtain the pH calibration curve. In each experiment, 20 cells were monitored per dish. Fiji program was applied for the quantification of images. pHi was calculated from the ratio of light intensities emitted at 535 nm after excitation at 440 and 490 nm. For the measurement of extracellular pH, bEnd.3 cells were cultured in 12-well plates and treated with 60 mM KCl for 5 minutes aCSF was applied as a control. Extracellular pH was measured before and at 5, and 30 minutes after the KCl pulse using FiveEasy Plus pH meter FP20 (Mettler Toledo) with Micro pH electrode LE422 over three individual experiments run in triplicate.

## Trans Endothelial Electrical Resistance (TEER)

TEER is an established method of evaluating endothelial cell barrier integrity which captures active and passive breaches *in vitro* [44]. bEnd.3 mouse endothelial cells were seeded at a density of $6.0 \times 10^4$ cells/cm$^2$ on the luminal side of collagen-coated trans-well inserts (Corning, 3460) with bEnd.3 media for growth facilitation at 37°C incubation. The abluminal side of inserts was treated with astrocyte conditioned media (ACM) at point of cell seeding to facilitate formation of tight junctions between cells of the endothelial monolayer. Baseline TEER measurements were obtained before each experiment. The abluminal side of trans-well insert was then treated with media (naïve group), pH (6.8–7.6), aCSF, or KCl (60 mM) for 5 minutes (KCl pulse). Pulse media was removed and replaced with fresh bEnd.3 media. TEER was assessed via 2-electrode, chopstick method (EVOM2) at the following timepoints immediately post-pulse, 10, 20, and 30 minutes after replacement of abluminal media. All measurements were repeated in triplicate over three individual experiments.

## *In vitro* uptake experiments

Functional implications of monolayer integrity or lack thereof were assessed *in vitro* as luminal to abluminal transport/uptake of select compounds. bEnd.3 cells were seeded at a density of $6.0 \times 10^4$ cells/cm$^2$ on the luminal side of collagen-coated filter membranes (0.4 μm pore polyester membrane) of 24-well tissue culture inserts (Costar, 3470). The tissue culture inserts were incubated at 37°C (5% $CO_2$) for 4–5 days. One day before experiment, astrocyte (C8-D1A cells)-conditioned media was added to the abluminal side of the inserts and incubated overnight in the humidified incubator at 37°C (5% $CO_2$). On the day of experiment, zoniporide (10 nM) was added to the luminal side of inserts 30 min before KCl pulse. $^{14}$C-sucrose (PerkinElmer, NEC100XOO1MC) was applied to the luminal side to monitor the paracellular uptake. In a separate set of experiments, $^3$H-sumatriptan (American RadioLabeled Chemicals, ART1619) was used to determine *in vitro* uptake of a clinically relevant compound. Both radiolabeled compounds were added to luminal side at 0.25 μCi/ml concentration. The radioactivity of samples from the abluminal side was measured for disintegrations per minute (dpm; 1450 LSC and Luminescence Counter; PerkinElmer) at 5 and 30 minutes after aCSF or KCl-pulse (60 mM, 5 min) (aCSF or KCl was added to the abluminal side when radioligands were applied). For each individual experiment, 3–4 inserts/group with cells and three inserts without cells were assayed. The radioactivity in the inserts without cells at both time-points were two-three times higher compared to the inserts with cells, indicating the existence of barrier. All experiments were carried out in triplicate, using 3–4 trans-well inserts/group.

### Design and cloning sgRNA for *SLC9A1* gene targeting

We identified and extracted the sequence of the *SLC9A1* gene from the genomic sequence of chromosome 4 (133.369.706–133.423.702) from *Mus musculus* using https[https://www.ensembl.org/](https://www.ensembl.org/). To design the gRNA sequence, we screened for potential off-targets using the gRNA design tool ([https://crispr.cos.uni-heidelberg.de](https://crispr.cos.uni-heidelberg.de)). The sgRNA selected for targeting *SLC9A1* exon 1 is located on the forward strand: ATCTTCCCCTCCTTGCTGG, score 86. pL-CRISPR.EFS.GFP (Addgene, 57818) was used as a backbone vector for simultaneous expression of Cas9 enzyme with the sgRNA and GFP to control for the infection efficiency. The plasmid was cut using FastDigest Esp3I (ThermoScientific, FD0454) according to manufacturer's instructions. The digested plasmid was extracted from 0.8% agarose gel using Gene-JET Gel extraction kit (ThermoScientific, K0691). Oligonucleotides purchased from IDT were annealed and phosphorylated using T4 polynucleotide kinase (New England BioLab, M0201S) in a thermocycler applying the following program: 37˚C for 30 min, 65˚C for 20 min. Then, 50 ng of the digested pL-CRISPR.EFS.GFP plasmid set up for ligation with 200 nM of the annealed oligonucleotides at 22˚C for 5 minutes using Rapid DNA Ligation kit (Thermo-Scientfic, K1422). The ligation products were transformed into NEB® Stable Competent *E. coli* bacteria (New England BioLab, c3040) according to manufacturer's instructions. Plasmids were purified using QIAprep Spin Miniprep Kit (Qiagen, 27104) and QIAGEN Plasmid Maxi Kit (Qiagen, 12162).

### Lentiviral infection of bEnd.3 cells

The lentiviral particles were produced in HEK293T cells (ATCC, CRL-3216) using PEI transfection method. The lentivirus was purified by Lenti-X™ Concentrator (Clontech, 631231) and tittered by qPCR Lentivirus Titration Kit (Applied Biological Materials Inc., LV900-S). bEnd.3 cells were plated in 6-well plates and allowed to adhere overnight. Virus was added at the indicated titer in the presence of polybrene (2μg/mL, Sigma, H9268) and the cells were incubated for 24 h. After 24 h, the media was changed to fresh media with no virus and the cells were incubated for additional 4–5 days, then subjected to the subsequent experiments (e.g., Western-blotting and *in vitro*-uptake experiments).

### Data analysis and statistics

GraphPad Prism 7.0 software (GraphPad Software) was used for statistical analysis. Unless otherwise stated, the data were expressed as mean ± SEM. Periorbital allodynia measurements were assessed using a repeated measure two-way ANOVA to analyze differences between treatment groups over time with either a Bonferroni or Tukey test applied *post hoc*. Molecular studies were compared by unpaired t-test or one-way ANOVA, as indicated. Differences were considered significant if $p \leq 0.05$ to give 80% power to detect at 20% difference and prevent a type II error (GPower3.1).

## Results

### Functional NHE1 is required for intact paracellular BBB integrity *in vitro*

Recently, NHE1 inhibition was reported as a viable strategy to alleviate BBB injury induced by stroke and subarachnoid hemorrhage in animal models [26–29,45]. We asked if functional expression of NHE1 is critical to maintaining paracellular integrity of the BBB using an *in vitro* model. bEnd.3 cells cultured in collagen-coated trans-well inserts in the presence of astrocyte-conditioned media formed a tight monolayer as evaluated by TEER (Fig 1A). Five-minute pulses on the abluminal side with KCl (60mM), but not aCSF significantly reduced TEER

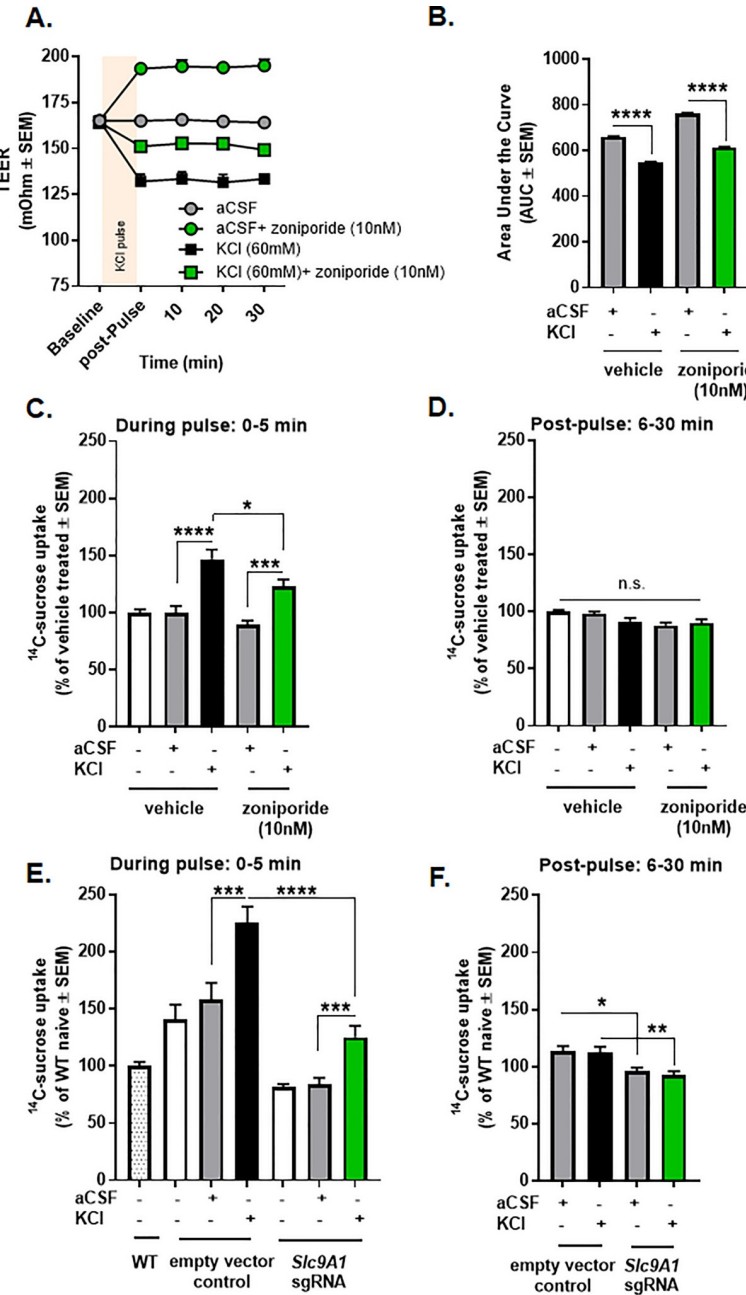

**Fig 1. KCl pulse induced temporary paracellular leak in bEnd.3 endothelial cells, which was reduced, but not completely abolished by selective inhibition of NHE1. (A)** TEER values after a 5 min pulse of KCl (60 mM) or aCSF with or without pretreatment of zoniporide (10 nM). Values are the mean ± SEM (n = 9). **(B)** The area under the curve for TEER values. **(C)** $^{14}$C-sucrose uptake during a 5 min KCl or aCSF pulse with or without pretreatment of zoniporide (10 nM). All data represent % of vehicle treated ± SEM (n = 9–11). **(D)** $^{14}$C-sucrose uptake 30 min post-pulse with or without pretreatment of zoniporide (10 nM). All data represent % of vehicle treated ± SEM (n = 9–11). **(E)** $^{14}$C-sucrose uptake during KCl or aCSF pulse after the genetic engineering of NHE1. All data represent % of wild-type naïve ± SEM (n = 9). **(F)** $^{14}$C-sucrose uptake 30 min post-pulse after the genetic engineering of NHE1. All data represent % of wild-type naïve ± SEM (n = 9). * $p < 0.05$, ** $p < 0.01$, *** $p < 0.001$, **** $p < 0.0001$ as assessed by one-way ANOVA. n.s. = non-significant.

values as compared to baseline values indicating a breach in endothelial monolayer integrity (Fig 1A and 1B) (AUC- aCSF: 660.5±2.957 vs. KCl: 546.5±5.048, p<0.0001, as assessed by one-way ANOVA with Tukey post-test, n = 9/group, F(3,32) = 4682). Reductions in TEER values paralleled those seen at pH 6.8,7, and 7.6 (S1 Fig).

The selective NHE1 inhibitor, zoniporide, was applied to aCSF- or KCl-treated cells to determine if NHE1 plays a role in KCl-mediated breaches in BBB integrity *in vitro*. Thirty-min pretreatment with zoniporide (10 nM) significantly increased TEER values in both conditions (Fig 1A and 1B) (AUC- aCSF: 660.5±2.957 vs. aCSF+zoniporide: 762.8.5±4.242, p<0.0001, as assessed by one-way ANOVA with Tukey post-test, n = 9/group; KCl: 546.5±5.048 vs. KCl+-zoniporide: 613.6±3.368, p<0.0001, as assessed by one-way ANOVA with Tukey post-test, n = 9/group, F(3,32) = 4682).

To determine if KCl-mediated reductions in TEER were associated with paracellular leak, we next assessed $^{14}$C-sucrose uptake after the KCl pulse. $^{14}$C-sucrose as a marker of BBB paracellular permeability was used in these studies in line with our former *in vivo* experiments, in which elevated cortical uptake of $^{14}$C-sucrose was observed after induction of CSD [36]. The uptake of $^{14}$C-sucrose through the monolayer of bEnd.3 cells were significantly increased during KCl-pulse (Fig 1C) (aCSF: 100.0±5.6% of vehicle-treated vs. KCl: 146.8±8.5% of vehicle-treated, p<0.0001, as assessed by one-way ANOVA with Tukey post-test, n = 9-11/group, F(4,48) = 16.08). No significant differences were observed in samples collected 30 min after aCSF or KCl pulses (Fig 1D) (aCSF: 98.2±1.8% of vehicle-treated vs. KCl: 91.5±3.1% of vehicle-treated, p = 0.34, as assessed by one-way ANOVA with Tukey post-test, n = 9-11/group), suggesting transient paracellular leak caused by KCl.

Pharmacological blockade of NHE1 by zoniporide did not influence $^{14}$C-sucrose uptake (Fig 1C) (aCSF: 100.0±5.6% of vehicle-treated vs. aCSF+zoniporide: 89.0±4.1% of vehicle-treated, p = 0.61, as assessed by one-way ANOVA with Tukey post-test, n = 11/group). In contrast, the inhibition of NHE1 by zoniporide attenuated, but did not completely abolish, the increased abluminal content of $^{14}$C-sucrose induced by KCl (Fig 1C) (KCl: 146.8±8.5% of vehicle-treated vs. KCl+zoniporide: 122.9±6.2% of vehicle-treated, p = 0.03, as assessed by one-way ANOVA with Tukey post-test, n = 9-11/group, F(4,48) = 16.08).

In order to exclude the possible off-target activity of pharmacological inhibition, we tested the genetic manipulation of NHE1. The efficacy of genetic manipulation of NHE1 was monitored by GFP-positive fluorescence and Western immunoblotting (Fig 2). The genetic knockdown of NHE1 showed similar effect on the uptake of $^{14}$C-sucrose (Fig 1E and 1F). The paracellular permeability of $^{14}$C-sucrose was significantly decreased in NHE1 knock-down cells compared to empty vector control during KCl pulse (Fig 1E) (empty vector control KCl: 225.6±13.9% of wild-type naive vs. *Slc9A1* sgRNA KCl: 125.1±10.1% of wild-type naive, p<0.001; *Slc9A1* sgRNA aCSF: 84.2±5.5% of wild-type naive vs. *Slc9A1* sgRNA KCl: 125.1±10.1% of wild-type naïve, p<0.001, as assessed by one-way ANOVA with Tukey post-test, n = 9/group, F(6,55) = 24.41). Significant differences between aCSF vs. KCl-treated uptake at 6–30 min post-pulse were not observed in gene edited cells, confirming the temporary effect of KCl on paracellular leak. However, the genetic manipulation of NHE1 revealed a long-lasting effect on BBB permeability, decreasing $^{14}$C-sucrose uptake post KCl-pulse (Fig 1F) (empty vector control aCSF: 113.9±4.1% of wild-type naive vs. *Slc9A1* sgRNA aCSF: 96.1±3.2% of wild-type naive, p = 0.02; empty vector control KCl: 112.2±5.3% of wild-type naive vs. *Slc9A1* sgRNA KCl: 92.7±3.4% of wild-type naive, p<0.01 as assessed by one-way ANOVA with Tukey post-test, n = 9/group, F(3,32) = 7.129). Together, these studies indicate that functional NHE1 expression is required for paracellular integrity of brain endothelial cells during high extracellular potassium *in vitro*.

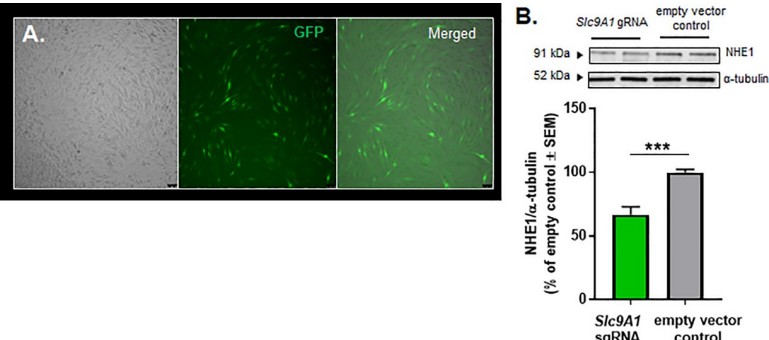

**Fig 2. Monitoring the efficacy of CRISPR-Cas9 system in bEnd.3 cells.** (**A**) Representative images of bEnd.3 cells infected with pL.CRISPR.EFS.GFP-*Slc9A1* sgRNA show GFP-positive cells 72 hours after post-infection. (**B**) Representative immunoblots indicating NHE1 and α-tubulin as a loading control in the whole cell lysate of gene-edited bEnd.3 cells. Values represent the % of empty vector control ± SEM (n = 9–10). *** p = 0.0002, as assessed by unpaired t-test.

## Loss of functional NHE1 activity facilitates sumatriptan uptake

In the next set of experiments, we wanted to examine the role of NHE1 on the abluminal uptake of antimigraine agents; $^3$H-sumatriptan was used as a model molecule. Similarly to $^{14}$C-sucrose uptake, the transport of $^3$H-sumatriptan across the bEnd.3 endothelial monolayer was enhanced during the KCl pulse (Fig 3A; aCSF: 115.3±7.2% of vehicle-treated vs. KCl: 142.2±6.2% of vehicle-treated, p = 0.01, one-way ANOVA, Tukey post-test, n = 18/group, F (4,76) = 26.98). No significant differences were observed 6–30 min post aCSF or KCl (Fig 3B; aCSF: 96.4±4.1% of vehicle-treated vs. KCl: 96.0±3.9% of vehicle-treated, p>0.99, one-way ANOVA with Tukey post-test, n = 11-13/group). Unexpectedly, inhibition of NHE1 with zoniporide significantly enhanced the abluminal uptake of $^3$H-sumatriptan during KCl-pulse (Fig 3A; KCl: 142.2±6.2% of vehicle-treated vs. KCl+zoniporide: 186.4±6.3% of vehicle-treated, p<0.0001, one-way ANOVA with Tukey post-test, n = 13/group, F(4,76) = 26.98). The uptake of $^3$H-sumatriptan during KCl pulse was further increased by NHE1 knock-down (Fig 3C; empty vector control KCl: 168.8±9.1% of wild-type naive vs. *Slc9A1* sgRNA KCl: 214.1±11.7% of wild-type naive, p<0.01, as assessed by one-way ANOVA with Tukey post-test, n = 9-13/group, F(6,69) = 24.45). These observations suggest that careful manipulation of pH via NHE1 may be a strategy to selectively enhance blood to CNS uptake of sumatriptan to increase therapeutic efficacy.

## NHE1 inhibition enhances sumatriptan efficacy after cortical injection of KCl *in vivo*

Prior to assessing NHE1 contribution *in vivo*, we investigated the induction of periorbital allodynia following cortical KCl injections in female rats as previously reported in male rats [36,37]. Cortical KCl (0.5μL 1M) but not aCSF (0.5μL) induced significant periorbital allodynia that lasted for >24h (Fig 4). To address whether periorbital allodynia in the KCl resulted from activation of trigeminal afferents and/or cortical excitation during the KCl injection, separate cohorts of rats were evaluated after dural application of KCl in combination with dural application of either saline or lidocaine (2%); facial sensitivity was not observed in these animals (Fig 4B). Tactile sensitivity was not prevented by dural application of lidocaine prior to the cortical KCl injection (Fig 4C).

To test whether pH manipulation via NHE1 inhibition is a useful strategy to enhance antiallodynic efficacy of sumatriptan following cortical KCl injections as suggested by our *in vitro*

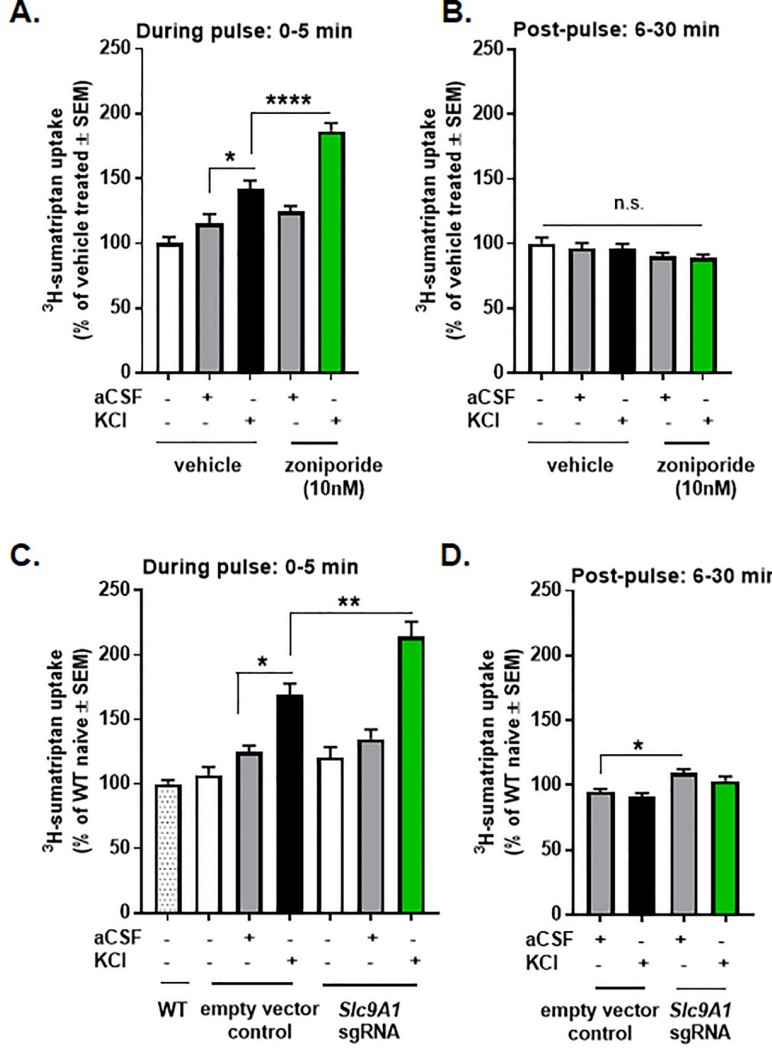

**Fig 3. NHE1 inhibition enhanced KCl induced transport of sumatriptan *in vitro*.** (**A**) [3]H-sumatriptan uptake during a 5 min KCl or aCSF pulse with or without pretreatment of zoniporide (10nM). All data represent % of vehicle treated ± SEM (n = 13–18). (**B**) [3]H-sumatriptan uptake post-pulse with or without pretreatment of zoniporide. All data represent % of vehicle treated ± SEM (n = 11–13). (**C**) [3]H-sumatriptan uptake during the KCl or aCSF pulse after the genetic manipulation of NHE1. All data represent % of wild-type naïve ± SEM (n = 9–13). (**D**) [3]H-sumatriptan uptake 30 min post-pulse after the genetic manipulation of NHE1. All data represent % of wild-type naïve ± SEM (n = 9–13). * p<0.05, ** p<0.01, **** p<0.0001 as assessed by one-way ANOVA. n.s. = non-significant.

results, rats were treated with zoniporide (5mg/kg IP,) or saline (1mL/kg, IP) 40 min before the cortical KCl injection (0.5μL, 1M, t = -30 min). After induction of periorbital allodynia, sumatriptan (0.6mg/kg, SC, t = 0 min) was administered (Fig 5A). Zoniporide pretreatment significantly enhanced sumatriptan reversal of cortical KCl induced periorbital allodynia 30 min after sumatriptan administration as compared to post-baseline (zoniporide+KCl+-sumatriptan pBL vs. 30 min, p = 0.03, two-way ANOVA with Tukey's post-test, n = 5-10/group, F(6,78) = 6.536) and saline pretreatment (zoniporide+KCl+sumatriptan 30 min vs. saline+KCl+sumatriptan 30 min, p = 0.03, two-way ANOVA with Bonferroni's post-test, n = 5-10/group, F(6,78) = 6.536) (Fig 5B).

Sumatriptan is known to have variable clinical responses [46]. We next assessed whether zoniporide pretreatment changed the number of sumatriptan sensitive animals. Sumatriptan

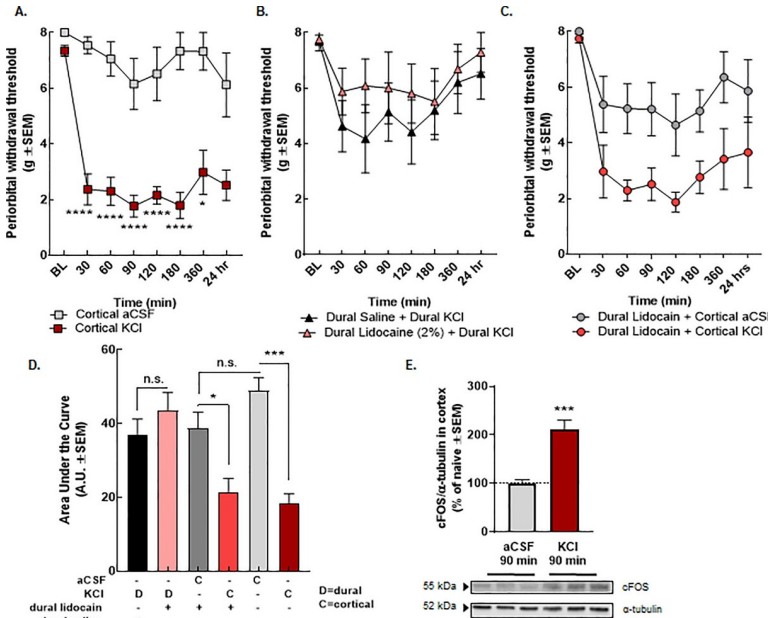

**Fig 4. Cortical KCl, but not dural injection, leads to periorbital allodynia in female rats and activates cFOS.** (**A-D**) Dural stimulation with KCl at up to 10x the volume (5uL, 1M) does not induce significant periorbital allodynia as compared to cortical KCl (0.5uL, 1M). Data are expressed mean ± SEM (n = 8–11), one or two-way ANOVA, *p<0.05,***p<0.001, ****p<0.0001. (**E**) The expression of cFOS in cortex was elevated at 90 min after cortical KCl. The dashed line indicates the relative expression of cFOS in naïve cortex. Data are expressed % of naive ± SEM (n = 3–4), one-way ANOVA, ***p<0.001.

responders were defined as having a >50% antiallodynic response for ≥ 2 timepoints within the first 120 min; non-responders had <50% anti-allodynia throughout the first 2 h. Of the saline-pretreated rats, 40% responded to sumatriptan whereas zoniporide pretreatment increased sumatriptan sensitivity to 85% of the animals (Fig 5C).

## Selective inhibition of NHE1 by zoniporide mimics KCl-induced facial sensitivity

A recent review paper discussed the role of NHE1 in nociception, drawing attention its possible protective role in pain models [47]. These studies and our observation that pretreatment with zoniporide did not fully prevent the induction of the KCl-mediated periorbital allodynia led to evaluation of the effect of zoniporide administration in the absence of sumatriptan after cortical aCSF or KCl injections; rats were treated with zoniporide (1 mg/kg, IP) at 10 min before cortical KCl or aCSF injection (Fig 5D). Pretreatment with zoniporide did not alter the facial allodynia induced by KCl (Fig 5E and 5F) (AUC- KCl: 11.96 ± 2.02 vs. KCl+zoniporide: 13.6 ± 3.13, p = 0.97, one-way ANOVA with Tukey post-test, n = 5-6/group, F(3,20) = 7.033). However, systemic administration of zoniporide to aCSF control rats induced facial sensitivity statistically indistinguishable from cortical KCl (AUC- KCl: 11.96 ± 2.02 vs. aCSF+zoniporide: 13.51 ± 2.04, p = 0.98, one-way ANOVA with Tukey post-test, n = 5-6/group, F(3,20) = 7.33). These data suggest that systemic NHE1 inhibition in the absence of injury may induce periorbital allodynia in female rats.

Previously we have shown that CSD events *in vivo* increase paracellular uptake of $^{14}$C-sucrose, but not Evans Blue albumin [36], in line with above findings. To further investigate the role of NHE1 in the maintenance of BBB integrity, we assessed the uptake of the medium

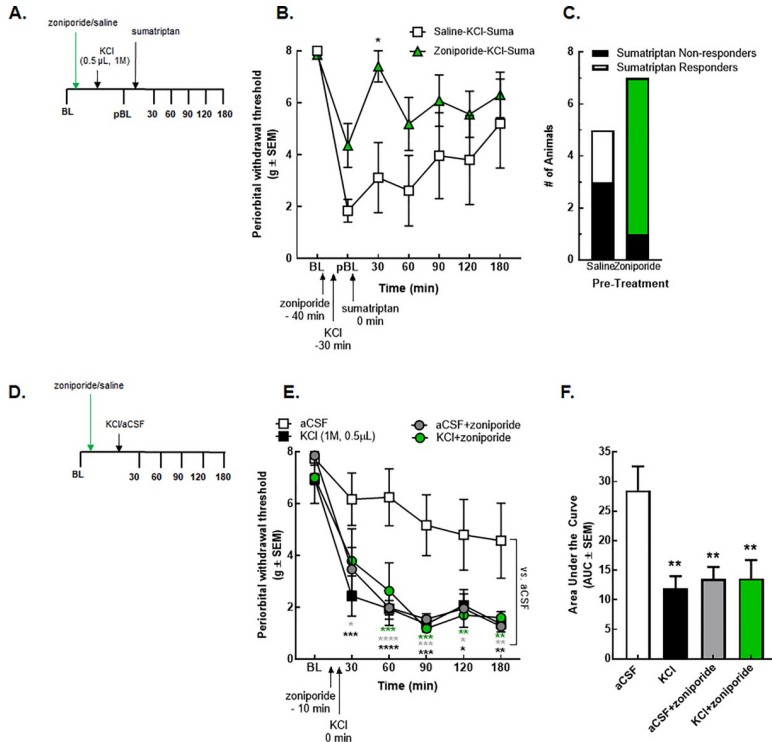

**Fig 5. Systemic inhibition of NHE1 by zoniporide influenced the overall efficacy of sumatriptan but induced periorbital allodynia alone.** (**A**) Timeline of the zoniporide-sumatriptan combination treatments. (**B**) Periorbital withdrawal threshold determined by von Frey test in rats treated with zoniporide (5 mg/kg, IP) or saline (1mL/kg, IP) in combination with sumatriptan (0.6 mg/kg, SC) and cortical KCl injection (0.5 µL, 1M). Values are the mean ± SEM (n = 5–10). * denotes significantly different (p<0.05) compared to post-baseline or saline-treated as assessed by two-way ANOVA. (**C**) Number of the sumatriptan responder and non-responder animals. (**D**) Timeline of the zoniporide-KCl/aCSF combination treatments. (**E**) Periorbital withdrawal threshold determined by von Frey test in rats injected with cortical KCl (0.5 µL, 1M) or aCSF (0.5 µL) with zoniporide (1 mg/kg, IP) or saline (1mL/kg, IP). Values are the mean ± SEM (n = 5–6). * p<0.05, ** p<0.01, *** p<0.001, and **** p<0.0001 compared to aCSF-treated as assessed by two-way ANOVA. (**F**) The area under the curve for the animals treated with aCSF/KCl ± zoniporide. Values are the mean ± SEM (n = 5–6). ** denotes significantly different (p<0.01) compared to aCSF-treated as assessed by one-way ANOVA.

sized FITC-dextran (4000 Da) after cortical injection of aCSF or KCl ± the NHE1 inhibitor, zoniporide. Parenchymal detection of FITC fluorescence 90 min after cortical injections was below the limit of detection (20 ng/mL) for all groups (naïve, saline-aCSF, saline-KCl, zoniporide-aCSF, and zoniporide-KCl, n = 3-6/condition). These results suggest that BBB breaches *in vivo* after cortical KCl injections do not allow passage of molecules ≥ 4000 g/mol. Under these conditions, making conclusions about the role of NHE1 on BBB integrity is limited to small molecule/peptide uptake.

## Cortical KCl injection alter the expression of NHE1 in discrete pain nuclei engaged in CSD event

Given the finding that NHE1 inhibition alone could increase facial sensitivity and the record of pH changes during cortical spreading depression [11–14], we next assessed whether NHE1 expression was reduced following cortical injection of KCl. The total amount of NHE1 protein in areas directly (i.e. occipital cortex, Ct) and indirectly (i.e. periaqueductal grey, PAG) affected by CSD [40,48] as well as discrete pain nuclei (trigeminal nucleus caudalis, VC; and trigeminal ganglia, TG) was assessed by Western immunoblotting 90 min after cortical aCSF or KCl.

In Ct samples, Western blot detection of total NHE1 was significantly decreased at 90 min after dural manipulation (Fig 6A; naïve: 101.1±4.5, aCSF: 111.1±11.7, KCl: 83.2±1.8; n = 3, loaded in duplicate, aCSF vs. KCl, p = 0.04, one-way ANOVA with Tukey post-test, $F_{(2,8)}$ = 2.800). Reduced expression of NHE1 was also observed in the PAG samples at the same time-point (Fig 6B; naïve: 100.0±10.9, aCSF: 85.6±7.9, KCl: 47.3±6.1; n = 3, loaded in duplicate, aCSF vs. KCl, p = 0.016, one-way ANOVA with Tukey post-test, $F_{(2,15)}$ = 10.14).

The expression of NHE1 in trigeminal nucleus caudalis (Vc) at 90 min after cortical KCl injection did not show significant changes (Fig 6C: Vc- naïve: 100.0±16.4, aCSF: 99.5±17.1, KCl: 79.3±13.7, n = 3, loaded in duplicate, aCSF vs. KCl p = 0.61; one-way ANOVA with Tukey post-test, $F_{(2,28)}$ = 0.495). However, the total NHE1 expression in trigeminal ganglia (TG) was elevated 90 min after induction of CSD (Fig 6D TG- naïve: 100.0±11.6, aCSF: 94.1±10.1, KCl: 148.3±18.2, n = 6, aCSF vs. KCl p = 0.035, one-way ANOVA with Tukey post-test $F_{(2,15)}$ = 4.669). These data indicate that the expression of NHE1 is impaired in regions engaged by CSD events but not primary pain neurotransmission 90 min after cortical KCl injection.

## The KCl pulse induced functional loss of NHE1 in bEnd.3 endothelial cells but no other cells of the NVU

Cortical KCl-induced CSD events cause transient, time-and regional dependent changes in BBB integrity [36] and functional NHE1 expression was altered in CSD tissue. It is well-known that the neurovascular unit (NVU) throughout the parenchyma is comprised of endo-thelial cells, pericytes, astrocytes, neurons, and microglia [30]. The next studies explored the cell-selective effect of high extracellular $K^+$ on NHE1 expression using *in vitro* methods. Following a KCl pulse (60mM, 5 min) or aCSF exposure, total expression of NHE1 in mouse brain endothelial cells (bEnd.3), astrocytes (C8-D1A), microglia (C8-B4), and primary culture of trigeminal ganglia was quantified by immunocytochemistry. Immunofluorescence images of bEnd.3 cells revealed internalization of NHE1 after KCl pulse in endothelial cells only, which was completely recovered by 30 min (Fig 7A). In contrast, we did not observe changes in the total expression or localization of NHE1 in other cell types (Fig 8).

Given the suggested change in NHE1 localization from ICC (Fig 7A), subcellular fraction-ation followed by semiquantitative Western blot was performed. The KCl (60mM, 5 min) pulse resulted in a significant increase in nuclear NHE1 as compared to membrane NHE1 (two-tailed Students' T-test p = 0.002; Fig 5B'), but not other compartmental ratios (membrane/cytosol (p = 0.92): Fig 7B" and cytosol/nucleus (p = 0.41): Fig 7B'''). The KCl treatment did not cause significant changes in the total expression of NHE1 at the 5 min time-point (aCSF = 100.0±3.7, KCl = 112.9±5.3; n = 5, ns = non-significant, as assessed by two-tailed t-test p>0.99; Fig 7C). NHE1 expression in the brain microvasculature was then evaluated in the whole lysate of corti-cal microvessels harvested 90 min after cortical KCl or aCSF injection. The immunoblots revealed no significant differences in the total expression of NHE1 between aCSF and KCl sam-ples, however NHE1 expression was significantly elevated after KCl treatment compared to naïve (naïve = 100.2±17.5, aCSF = 192.9±101.7, KCl = 343.4±177.6; n = 6, aCSF vs. KCl, p = 0.06; naïve vs. KCl, p = 0.005, as assessed by one-way ANOVA; $F_{(2,19)}$ = 6.884; Fig 7D). Our *in vitro* results indicate that the KCl pulse caused movement of NHE1 from the membrane to the nucleus in cell-specific manner during events when extracellular $K^+$ is high.

## KCl pulse caused intracellular acidosis in bEnd.3 endothelial cells but not extracellular acidosis

To test if observed changes in the localization of NHE1 were a coupled function, pH flux in bEnd.3 cells was determined. Intracellular pH (pHi) was assessed by ratiometric analysis of the

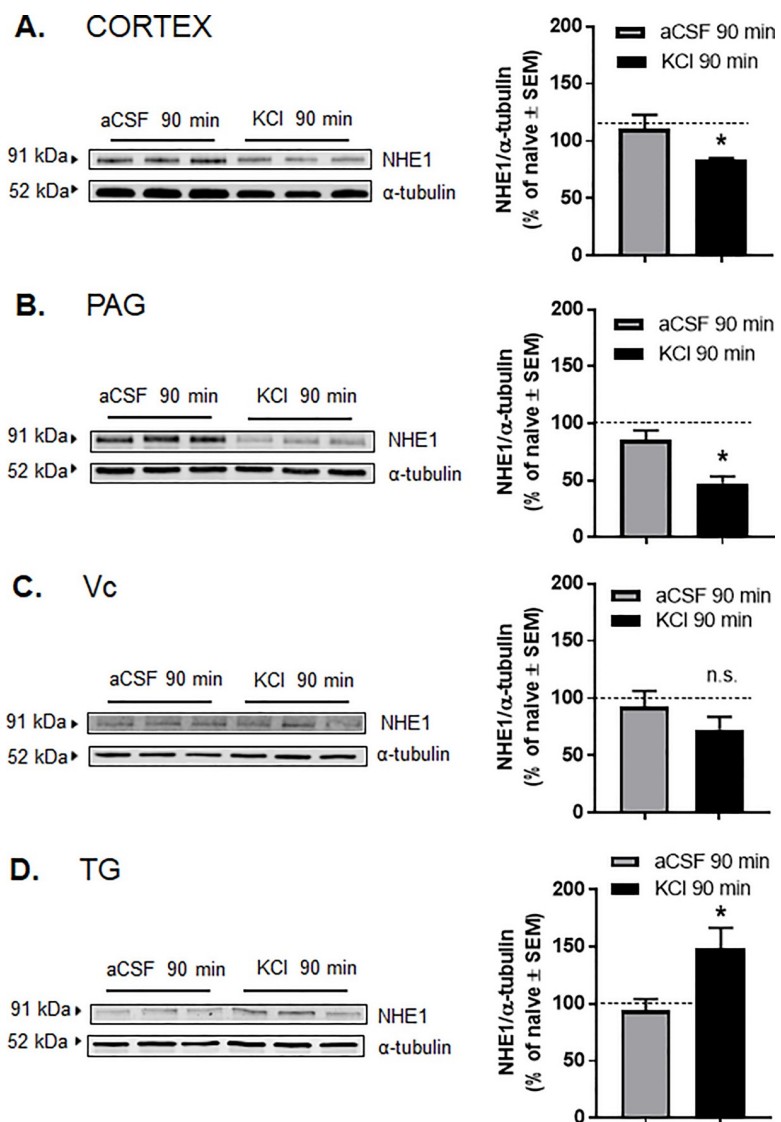

**Fig 6. Total expression of NHE1 was decreased in cortex and PAG samples at 90 min after dural manipulation.**
(A-D) Representative immunoblots indicating NHE1 and α-tubulin as a loading control in pain nuclei engaged in spreading depression, namely cortex (**A**), PAG (**B**), Vc (**C**), and TG (**D**). Samples were harvested 90 min after cortical aCSF or KCl injections. All data represent the % of naïve relative expression ± SEM (n = 3, loaded in duplicate). The dashed line indicates the relative expression of NHE1 in naïve samples. No significant differences were observed between naïve and aCSF-treated samples. * denotes significantly different (p<0.05), as assessed by one-way ANOVA.

pH-sensitive dye, BCECF-AM (5μM, 30 min loading), after successive application of media, aCSF or KCl (60mM) for 5 min each (Fig 7E). Intracellular pH did not significantly change during application of either media or aCSF (media: 6.97±0.15–7.37±0.28, p>0.99; aCSF: 6.94 ±0.07–7.32±0.11, p>0.99). Exposure to KCl (60mM, 5min) significantly reduced pHi of bEnd.3 endothelial cells from pH = 7.27±0.08 to 6.55±0.16 over 5 min (F(10,467) = 4.912; p = 0.0009 vs. time = 0min, two-way ANOVA with Bonferroni post-hoc). When these KCl-induced fluxes were compared to media and aCSF exposure, KCl significantly increased pHi after 3 min of incubation (p = 0.007 vs. media, p = 0.01 vs. aCSF, as assessed by two-way ANOVA, F(10,467) = 4.912) followed by significant acidification at 5 min post pulse

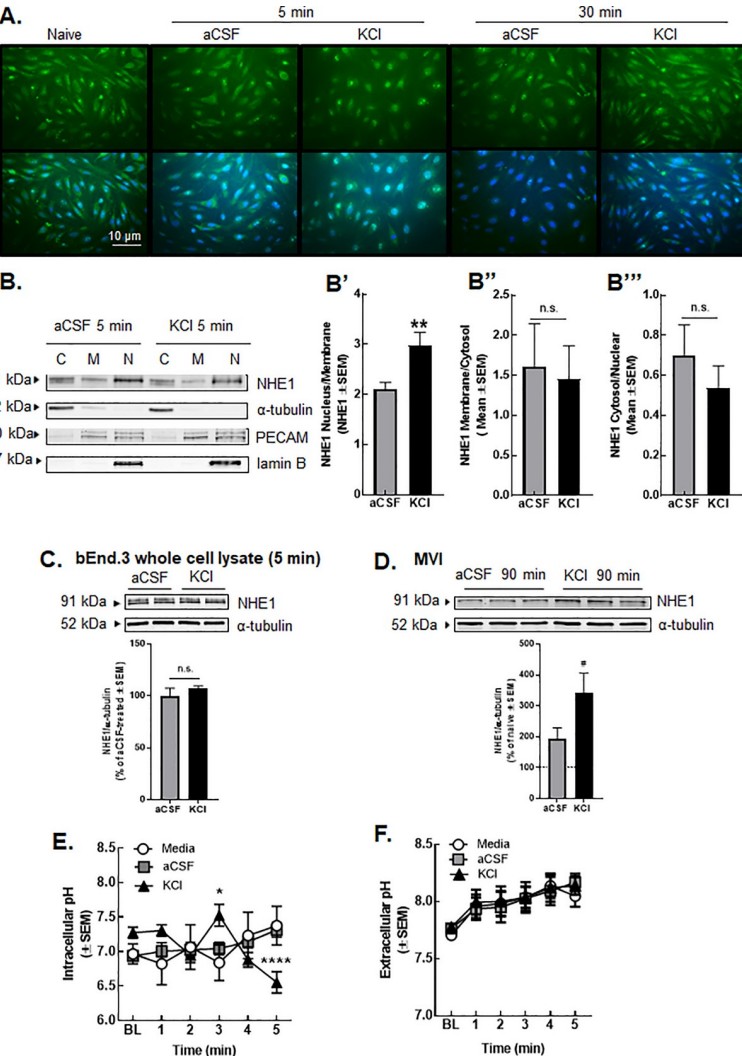

**Fig 7. KCl pulse decreased membrane, but increased nuclear, detection of NHE1 in b.End3 endothelial cells.** (**A**) Representative immunofluorescence images of bEnd.3 cells at two time-points after KCl or aCSF pulse. (**B**) Representative immunoblots of NHE1, α-tubulin, PECAM, and lamin B in cytosol, membrane, and nuclear fractions of bEnd.3 cells harvested at 5 min after KCl or aCSF pulse. (C = cytosol, M = membrane, N = nuclear) Values represent the mean ratio of NHE1 detection ± SEM (n = 11–12). (**C**) Representative immunoblots indicating NHE1 and α-tubulin as a loading control in whole cell lysate of bEnd.3 cells harvested at 5 min after KCl or aCSF pulse. Values represent the % of aCSF-treated relative expression ± SEM (n = 6). (**D**) Representative immunoblots of NHE1 and α-tubulin as a loading control in whole lysate of microvessels harvested at 90 min after cortical injection of KCl or aCSF. Values represent the % of naive relative expression ± SEM (n = 6). # denotes significantly different vs naïve (p<0.01), as assessed by one-way ANOVA (**E**) Intracellular pH during aCSF or KCl pulse. All data represent mean ± SEM (n = 45). (**F**) Extracellular pH during aCSF or KCl pulse. All data represent mean ± SEM (n = 6) in triplicate.

(p = 0.0009 vs. media, p<0.0001 vs. aCSF, as assessed by two-way ANOVA, F(10,467) = 4.912). These data suggest a rapid biphasic shift in intracellular pH of bEnd.3 cells when exposed within 5 min of high extracellular K$^+$.

If removal of NHE1 from the cell surface underscores the biphasic changes in pHi, an increase in extracellular pH may be expected. Using a mini pH probe, extracellular pH was assessed during the 5 min KCl pulse and 30 min after. Extracellular pH in media exposed cells increased in a linear manner over time; neither the extracellular pH of aCSF nor KCl treated cells was significantly different than media treated cells (Fig 7F, p>0.99 two-way ANOVA,

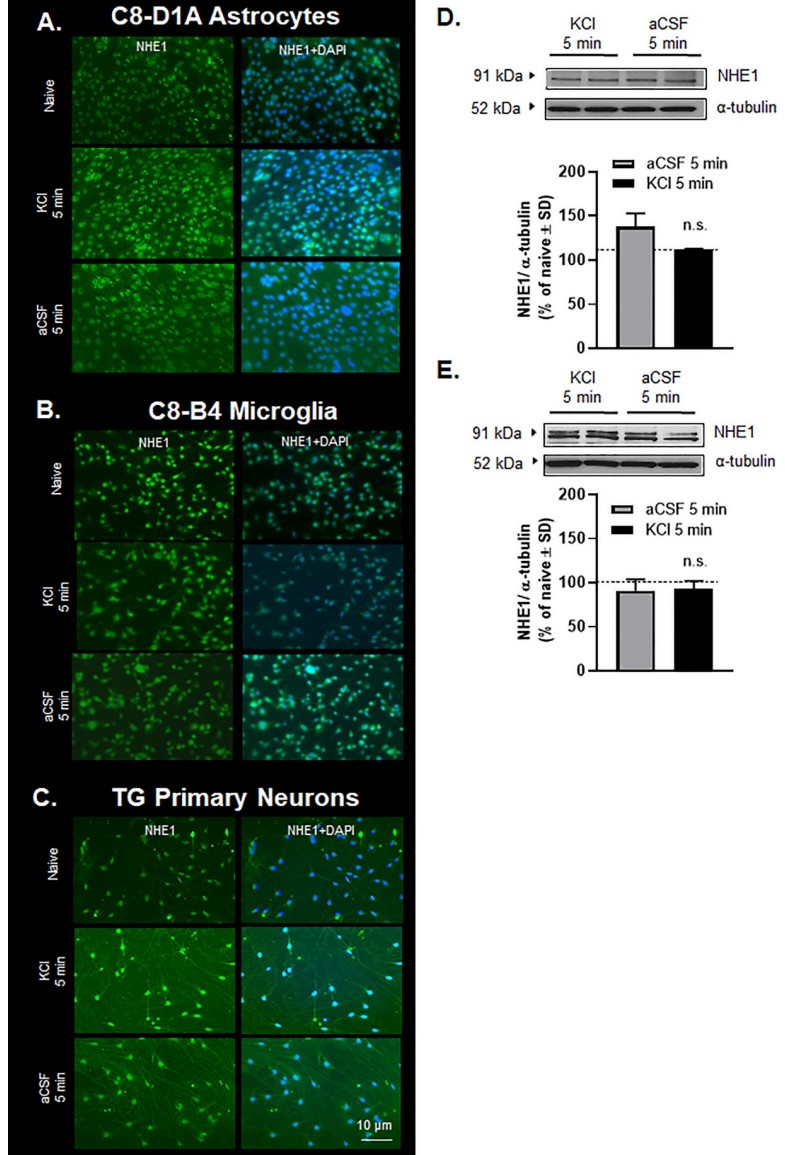

**Fig 8. No significant differences were observed in the localization or the total expression of NHE1 in astrocytes (C8-D1A), microglia (C8-B4), and primary culture of TG ganglion.** (A-C) Representative immunofluorescence images of astrocytes (C8-D1A), microglia (C8-B4), and primary culture of TG ganglion. (D) Representative immunoblots indicating NHE1 and α-tubulin as a loading control in the whole cell lysate of astrocytes. Values represent the % of naive relative expression ± SD (n = 2). (E) Representative immunoblots indicating NHE1 and α-tubulin as a loading control in the whole cell lysate of microglia. Values represent the % of naïve relative expression ± SD (n = 2). The dashed line indicates the relative expression of NHE1 in naïve samples. n.s. = non-significant.

Tukey post-hoc). Direct inhibition of NHE1 with zoniporide increased extracellular pH of bEnd.3 cells to 8.02±0.023 and 8.03±0.025 prior to aCSF and KCl, respectively; post-pulse pHe was not different between treatments (S2 Fig). Taken all together, the above data suggest that KCl (60mM, 5min) application to bEnd.3 brain endothelial cells induced a functional shift in pHi that may be attributed to movement of NHE1 from the membrane to the nucleus.

## Discussion

Migraine is one of the most prevalent neurological disorders, listed as the sixth most disabling disorder globally and the most incapacitating of all neurological disorders by the World Health Organization [49]. Approximately 90% of migraineurs experience moderate or severe pain, three quarters have a reduced ability to function during attacks leading to a high personal and socio-economic cost [50,51]. Moreover, 30% of migraineurs experience a phenomenon independent form headache pain called aura.

Cortical spreading depression has been considered as the underlying mechanism of the migraine aura [52] and is implicated in a number of other CNS disorders including traumatic brain injury, ischemic stroke, and hemorrhage [53–55]. Vascular dilation, depolarization of nociceptive afferents [56,57] and development of tactile allodynia [58–61] are linked to CSD events. In addition to behavioral, metabolic and blood flow changes, CSD events cause a dramatic failure of brain ion and pH homeostasis [11–14,62]. pH is a known regulator of BBB integrity that has documented effects on xenobiotic uptake [33,63–66]. The identity of which protein(s) regulate these transient barrier openings and pH changes during CSD events are unknown.

The sodium-hydrogen antiporter protein family is primarily responsible for maintaining the homeostasis of intracellular pH in the CNS [24,25,67–69]. Former literature data suggest that inhibition of one of the antiporters, NHE1, exhibits therapeutic potential, reducing BBB breakdown and neurovascular damage after acute ischemic stroke in animal models and cell-based systems [26–29]. Both pharmacological and genetic manipulation of NHE1 mitigated the sucrose permeability induced by KCl exposure *in vitro*, giving the first-line observation that NHE1 contributes to paracellular integrity of the BBB under these conditions.

In contrast to sucrose uptake, NHE1 inhibition and genetic knock down facilitated sumatriptan movement across bEnd.3 monolayers early, but not late, after KCl exposure suggesting an alternate method of uptake for sumatriptan across the blood brain barrier during a pathological state. *In vivo*, zoniporide pretreatment enhanced the onset of sumatriptan anti-allodynic effect, further supporting a role for pH regulation in therapeutic efficacy. Interestingly, the number of responders to sumatriptan was also increased in the zoniporide-treated group whereas the control group reflected the clinical percentage of sumatriptan sensitive patients [46], driving the intrigue that clinical variability in sumatriptan sensitivity may reflect NHE1 expression and/or activity. In fact, transient differences were observed in the efficacy of abortive drugs based on the phase of migraine in which they were administered, suggesting changes in BBB integrity during migraine [70]. Some clinical reports about MRI and CT scans of migraine patients and animal models of spreading depression further support the hypothesis of compromised BBB integrity during migraine attack [63–66,71,72].

The reduction of NHE1 functional expression seems to be beneficial to conserve paracellular BBB integrity, however, the data showing facial sensitivity induced by a single, systemic dose of zoniporide raises question about the overall positive effect of NHE1 blockade in migraine/secondary headache patients. The role of NHE1 in nociception has been recently discovered, suggesting a protective role of NHE1 in acute and chronic inflammatory pain [47]. Our results correspond well with the former data published showing that blockade of peripheral, as well as spinal, NHE1 by administration of selective and non-selective inhibitors, such as zoniporide, amiloride, and 5-(N,N-dimethyl)-amiloride increased pain behavior in capsaicin, serotonin, and formalin tests [73–76]. In line with these experiments, electrophysiological studies showed that the blockade of NHE1 can modulate electrical activity of primary nociceptive terminals [47].

A direct connection between NHE1 expression/activity and periorbital allodynia after cortical KCl injection was supported by these behavioral data. Significant decreases in the total expression of NHE1 were observed in cortex and PAG at 90 min, after injection of KCl, suggesting areas influenced by CSD events and implicated central sensitization have tightly regulated pH responses to depolarization [77–81]. It is well known that the brain function, including neuronal excitability and neurotransmitter release is highly sensitive to small changes in pH due to the high number of pH-sensitive membrane proteins, such as channels, transporters, receptors and ATPase pumps expressed in CNS [82,83].

In contrast to the Ct and PAG, the total expression of NHE1 was significantly elevated in the trigeminal primary afferents (TG) and it did not change at the first synapse (Vc). Interestingly, significant reduction of NHE1 protein level was observed in dorsal horn and DRG samples in formalin-induced inflammatory pain model [74], indicating regulation of NEH1 expression may be model or site specific in the context of pain. Future evaluations detailing the expression pattern of NHE1 in the CNS are warranted. Lastly, while periorbital allodynia in the KCl cohort of rats may reflect dual activation of trigeminal afferents and/or cortical excitation during the KCl injection (Fig 4); this was not observed in our initial studies [37]. These differences may be attributed to the volumes applied to the dura, sex, age, or other variables. However, the zoniporide/aCSF rats did not have chemically mediated dural stimulation suggesting that NHE1 function, rather than expression, may be altered in the trigeminal nociceptive circuit under these conditions. Future studies evaluating phosphorylation state and ion flux in these CNS regions will be more telling of the exact role of NHE1 in trigeminal pains.

NHE1 is expressed on most cells within the CNS [24,25,68] and CSD events engage any CNS component in the path [53,54,77]. To determine in which cells NHE1 expression and function negatively impacted *in vivo*, KCl pulse (60mM, 5 min) was applied to endothelial, microglial, astrocytic, and neurons *in vitro*, at extracellular $[K^+]$ (30 to 60 mmol/L) associated with CSD events [40,55]. Using complimentary techniques of immunocytochemistry, Western blot and subcellular protein fractionation, KCl application caused changes in NHE1 localization in endothelial cells (bEnd.3), suggesting their active participation in CSD events when extracellular $K^+$ is high. In the subcellular fractionation experiments, the presence of NHE1 in nuclear fraction was not expected. However, Bkaily et al. verified that NHE1 is present in nuclear membranes of cells known to express the antiporter using both immunofluorescence and confocal microscopy [84]. The quantification of intracellular pH in bEnd.3 cells after KCl pulse exhibited the functional consequences of the internalization of NHE1 under the same conditions revealing that brief excitation due to high extracellular $K^+$ is enough to change intracellular pH in the endothelial cells. We did not observe KCl-induced changes in the localization of NHE1 in other cell types available to us, including astrocytes, microglia, and primary culture of trigeminal ganglia. The observed reduction of NHE1 expression in tissue samples indicates contribution of other cell types of the CNS that may play a role, which requires further investigation. For example, pericytes were not investigated in the current study given the challenges in fully distinguishing this cell type from others (vascular smooth muscle cells) in immortalized lines and primary culture in the CNS [30,85–87]. Overall, our *in vitro* data suggest the active participation of endothelial cells in NHE1 regulation during periods when extracellular $K^+$ is high, including during cortical spreading depression events.

## Conclusions

It is well known that changes in pH can influence vascular tone (i.e., intracellular acidosis is associated with vasodilation) [88]. Several experimental papers provided evidence for a vascular-neuronal cross talk, a bi-directional communication between endothelium and neurons.

Moreover, Mason and Russo, in their review article suggest a new theory of vascular activation of nervous system underlying migraine pathophysiology [89]. The role of NHE1 in the development of various diseases of brain [69], such as cerebral ischemia-induced neuronal injury has been defined, but our work has demonstrated for the first time that NHE1 can modulate pain processing underlying headache like pain in rats, with a special emphasis of endothelial function and blood-brain barrier integrity. The involvement of other NHEs, like NHE3 and NHE5 in cortical spreading depression needs to be investigated in the future. Finally, the protective role of NHE1 suggest that normalization of pH homeostasis by targeting NHE1 function or surface expression could be a unique therapeutic approach to enhance efficacy of existing therapeutics for migraine.

## Supporting information

**S1 Fig. Reduction in TEER values was observed at extracellular pH 6.8, 7.0, and 7.6.** Nonlinear regression curve for TEER values at different extracellular pH at 5 and 30 min after the 5 min pH pulse. Values are % of baseline ± SEM (n = 3).
(TIF)

**S2 Fig. Post-pulse pHe was not different between aCSF vs. KCl treatment after direct inhibition of NHE1.** Direct inhibition of NHE1 with zoniporide increased extracellular pH of bEnd.3 cells to prior to aCSF and KCl. No significant difference between aCSF and KCl-treatment at any time-point was observed in these pre-treated cells. Values are mean ± SEM (n = 6). **** $p < 0.0001$ vs. baseline (BL), as assessed by two-way ANOVA with Tukey post-test.
(TIF)

**S1 Raw data.**
(PDF)

## Acknowledgments

The authors would like to acknowledge Patricia Jansma in the Imaging Core Marley at the University of Arizona for her assistance in collecting live cell images and the Delamere and Porreca labs for use of their LiCor developer.

## Author Contributions

**Conceptualization:** Erika Liktor-Busa, Tally M. Largent-Milnes.

**Data curation:** Erika Liktor-Busa, Kiera T. Blawn, Kathryn L. Kellohen, Tally M. Largent-Milnes.

**Formal analysis:** Erika Liktor-Busa, Kiera T. Blawn, Kathryn L. Kellohen, Vani Verkhovsky, Jared Wahl, Anjali Vivek, Seph M. Palomino, Tally M. Largent-Milnes.

**Funding acquisition:** Tally M. Largent-Milnes.

**Investigation:** Erika Liktor-Busa, Kiera T. Blawn, Kathryn L. Kellohen, Beth M. Wiese, Vani Verkhovsky, Jared Wahl, Anjali Vivek, Seph M. Palomino, Tally M. Largent-Milnes.

**Methodology:** Erika Liktor-Busa.

**Project administration:** Erika Liktor-Busa, Tally M. Largent-Milnes.

**Resources:** Tally M. Largent-Milnes.

**Supervision:** Erika Liktor-Busa, Tally M. Largent-Milnes.

**Writing – original draft:** Erika Liktor-Busa, Todd W. Vanderah, Tally M. Largent-Milnes.

**Writing – review & editing:** Erika Liktor-Busa, Thomas P. Davis, Tally M. Largent-Milnes.

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
