## [Decision Letter · Decision Letter 0]

20 Jan 2020

PONE-D-19-34794

Functional NHE1 Expression is Critical to Blood Brain Barrier Integrity and Sumatriptan Blood to Brain Uptake

PLOS ONE

Dear Largent,

Thank you for submitting your manuscript to PLOS ONE. After careful consideration, we feel that it has merit but does not fully meet PLOS ONE’s publication criteria as it currently stands. Therefore, we invite you to submit a revised version of the manuscript that addresses the points raised during the review process.

We would appreciate receiving your revised manuscript by Mar 05 2020 11:59PM. To enhance the reproducibility of your results, we recommend that if applicable you deposit your laboratory protocols in protocols.io, where a protocol can be assigned its own identifier (DOI) such that it can be cited independently in the future. For instructions see: http://journals.plos.org/plosone/s/submission-guidelines#loc-laboratory-protocols

We look forward to receiving your revised manuscript.

Kind regards,

Eliseo A Eugenin, Ph.D.

Academic Editor

PLOS ONE

Additional Editor Comments (if provided):

Dear Dr. Largent

Thank you for submit your manuscript to PLOSone. The reviewers had positive comments about your work and suggest some changes that need to be included. Please send your manuscript back with the changes requested

Best Regards

Eliseo

Journal Requirements:

Reviewers' comments:

Reviewer's Responses to Questions

**Comments to the Author**

1. Is the manuscript technically sound, and do the data support the conclusions?

Reviewer #1: Yes

Reviewer #2: Yes

2. Has the statistical analysis been performed appropriately and rigorously? 

Reviewer #1: Yes

Reviewer #2: Yes

3. Have the authors made all data underlying the findings in their manuscript fully available?

Reviewer #1: Yes

Reviewer #2: Yes

4. Is the manuscript presented in an intelligible fashion and written in standard English?

Reviewer #1: Yes

Reviewer #2: Yes

5. Review Comments to the Author

Reviewer #1: This is an interesting work evaluating the mechanisms of cortical spreading depolarization, a phenomenon linked to migraines. The work is solid and provides a number of significant insights that will be of interest to a broad neuroscience community. I have a few comments that require attention prior to further consideration of the work for publication in PLOS One.

a) The following statement outlining the overall goal of the study “Given that pH is a known regulator of 1) BBB integrity [30-32] and 2) transport of xenobiotics across the BBB [33,34] and that CSD induces regional perturbations in pH [35], the overall goals of the studies herein focused on NHE1 as a molecular target implicated in cortical KCl-induced deficiencies in BBB integrity resulting from pH dysfunction and regulation of sumatriptan blood to brain uptake” is challenging to parse out. It would be helpful to clarify in series the specific hypothesis of the paper.

b) As I understood it, the work is primarily focused on the role of NHE1 on the BBB integrity. However, the bulk of the mechanistic observations are made using cell culture assays, not in freshly dissected intact tissue or in vivo. There is a certain value of these model systems. In particular, induction of depolarization with KCl pulse along with a sucrose assay in endothelial cell monolayers is robust and reproducible. Having said that, this simplistic model system fails to replicate the precise architecture of the endothelial/pericyte system of the intact BBB and how it may be affected in the pathology. This limitation is my major concern and at least should be acknowledged and addressed in the Discussion. At best, the authors may want to address this by adding new experiment within a section starting on line 420 by testing the BBB integrity in vivo with either live Evan Blue assay or postmortem evaluation of plasma albumin, following the induction of CSD with KCl. I understand that these are extensive and technically challenging experiments that may be outside the scope of this work. Nevertheless, these would establish a direct link between CSD –BBB failure – a therapeutic action of NHE1 blockade.

c) Not clear the meaning of the lines 87-88: “…highlights that comprise of NHE1 may have differential effects on BBB 88 integrity versus xenobiotic uptake.”

d) Lines 357. Please provide a premise for using a sucrose uptake assay to test BBB integrity in endothelial cell monolayer.

e) The use of both pharmacological and genetic targeting of NHE1 and evaluation of pain are strength.

Reviewer #2: The manuscript, “Functional NHE1 Expression is Critical to Blood-Brain Barrier Integrity and Sumatriptan Blood to Brain Uptake”, Liktor-Busa et al. describe the dynamic role of sodium-hydrogen exchanger type 1 (NHE1) in blood-brain barrier (BBB) 31 integrity during cortical spreading depression events and the contributions of this antiporter on xenobiotic uptake. This is very interesting work and clinically relevant. The main strength of this study is that it is technically strong. However, I have a minor concern on the expression of NHE1 in Figure 4C and 4D. It would be in a better readable form if the authors replace these western blotting figures. Another concern is that the authors have given 4 supplementary figures. If the journal has an option to give these figures as main figures, it would be better to include as main figures.

6. PLOS authors have the option to publish the peer review history of their article (what does this mean?). If published, this will include your full peer review and any attached files.

Reviewer #1: Yes: Botir T Sagdullaev

Reviewer #2: No

---

## [Author Response · Author response to Decision Letter 0]

4 Mar 2020

Re: Revision of PONE-D-19-34794

Title: Functional NHE1 Expression is Critical to Blood Brain Barrier Integrity and Sumatriptan Blood to Brain Uptake

Response to Referees: The authors wish to thank the reviewers for their remarks and support for the research/manuscript. Below we have addressed individual concerns/comments in bold. Italicized points are authors’ interpretation of key reviewer concerns. Within the text of the manuscript, we have highlighted revision in red font. 

Reviewer #1: 

A) The following statement outlining the overall goal of the study “Given that pH is a known regulator of 1) BBB integrity [30-32] and 2) transport of xenobiotics across the BBB [33,34] and that CSD induces regional perturbations in pH [35], the overall goals of the studies herein focused on NHE1 as a molecular target implicated in cortical KCl-induced deficiencies in BBB integrity resulting from pH dysfunction and regulation of sumatriptan blood to brain uptake” is challenging to parse out. It would be helpful to clarify in series the specific hypothesis of the paper.

Answer: As the reviewer suggested, we clarified the specific hypothesis in the revised version of the paper.

B) As I understood it, the work is primarily focused on the role of NHE1 on the BBB integrity. However, the bulk of the mechanistic observations are made using cell culture assays, not in freshly dissected intact tissue or in vivo. There is a certain value of these model systems. Induction of depolarization with KCl pulse along with a sucrose assay in endothelial cell monolayers is robust and reproducible. Having said that, this simplistic model system fails to replicate the precise architecture of the endothelial/pericyte system of the intact BBB and how it may be affected in the pathology. 1) This limitation is my major concern and at least should be acknowledged and addressed in the Discussion. At best, the authors may want to address this by 2) adding new experiment within a section starting on line 420 by testing the BBB integrity in vivo with either live Evan Blue assay or postmortem evaluation of plasma albumin, following the induction of CSD with KCl. I understand that these are extensive and technically challenging experiments that may be outside the scope of this work. Nevertheless, these would establish a direct link between CSD –BBB failure – a therapeutic action of NHE1 blockade.

Answers:

B1) We agree with the reviewer that in vitro model systems of BBB are not able to fully replicate the precise architecture of the intact BBB. Despite their limitations, several different forms of cell-based systems are widely used and accepted model of BBB. In the current study, we used endothelial cells cultured in astrocyte-conditioned media in order to promote endothelial growth, differentiation and mimic the in vivo neurovascular unit. Our former data (Cottier et al. Loss of Blood-Brain Barrier Integrity in a KCl-Induced Model of Episodic Headache Enhances CNS Drug Delivery. eNeuro 2018; 5:ENEURO.0116-18.2018.) indicated that cortical KCl injection induced paracellular BBB leakiness to radiolabeled sucrose in the cortex, as assessed by in situ brain perfusion. We also observed an increase in BBB permeability to sumatriptan in both the cortex as well as the brainstem. The elevated permeability of sucrose and sumatriptan, observed in those experiment corresponded well with our in vitro data, presented in the current manuscript. 

B2) We appreciate the recommendation of the reviewer for “adding new experiment within a section starting on line 420 by testing the BBB integrity in vivo with either live Evan Blue assay or postmortem evaluation of plasma albumin, following the induction of CSD with KCl”. We have previously found in Cottier et al., citation above, that paracellular permeability to sucrose was detected. The ringer in this experiment contained Evan’s Blue Albumin; parenchymal uptake of EBA was not detected after cortical KCl injection to induce CSD. To delineate the link between CSD-BBB failure and NHE1 blockade as requested, we investigated FITC-dextran (4000 Da) transport in aCSF and KCl injected rats in the presence of either vehicle or zoniporide. Tissue was harvested 90 after cortical injection for consistency with other data in the current submission as well as Cottier et al., 2018. Parenchymal detection of FITC fluorescence in tissue samples was below the limit of detection (20 ng/mL) for all groups (naïve, saline-aCSF, saline-KCl, zoniporide-aCSF, and zoniporide-KCl), which suggest that BBB breaches in vivo after cortical KCl injections do not allow passage of molecules larger than 4kDa (please see method at line 149, results at line 535). 

Moreover we investigated the expression level of NHE1 in cortical microvessels, isolated from naïve, aCSF, and KCl-treated animals (see in Fig 7D). We did not observe significant differences between aCSF and KCl-treated animals, but the total expression of NHE1 was significantly increased 90 min after cortical KCl injection compared to naïve. 

C) Not clear the meaning of the lines 87-88: “…highlights that comprise of NHE1 may have differential effects on BBB 88 integrity versus xenobiotic uptake.”

Answer: We refined this sentence in the revised version. 

D) Lines 357. Please provide a premise for using a sucrose uptake assay to test BBB integrity in endothelial cell monolayer.

Answer: As mentioned above, radiolabeled sucrose was used in our former, in situ brain perfusion study (Cottier et al., 2018) to investigate changes in BBB paracellular permeability after CSD induction. Sucrose was applied in those and the current study as well, as a model molecule without known transcellular transporter. Therefore, it can represent the paracellular transport, and the paracellular leakage of BBB. We stated the rationale of using sucrose in the revised version of the manuscript. 

Reviewer #2: 

A) “However, I have a minor concern on the expression of NHE1 in Figure 4C and 4D. It would be in a better readable form if the authors replace these western blotting figures.”

Answer: As the reviewer suggested we replaced the representative Western blots within Fig 6C and 6D (Original Fig. number is 4, new number is 6). We would like to note that after quantification of a new set of TG samples, the expression of NHE1 after KCl treatment was significantly increased, as we indicated in the graph.

B) “Another concern is that the authors have given 4 supplementary figures. If the journal has an option to give these figures as main figures, it would be better to include as main figures.”

Answer: Having consulted the editorial office on the policy for Supplemental Information, we have moved the former S2, S3, and S4 figures to main figures and left S1 and S5 as supplemental figures.

We would like to note that our blot/gel image data with other raw data can be found in Supporting Information file.

---

## [Decision Letter · Decision Letter 1]

5 May 2020

Functional NHE1 Expression is Critical to Blood Brain Barrier Integrity and Sumatriptan Blood to Brain Uptake

PONE-D-19-34794R1

Dear Dr. Largent-Milnes

We are pleased to inform you that your manuscript has been judged scientifically suitable for publication and will be formally accepted for publication once it complies with all outstanding technical requirements.

With kind regards,

Eliseo A Eugenin, Ph.D.

Academic Editor

PLOS ONE

Additional Editor Comments (optional):

Thank you for a great re-submission and for taking the time and effort to answer all the questions and concerns

Best regards

Eliseo

Reviewers' comments:

Reviewer's Responses to Questions

**Comments to the Author**

1. If the authors have adequately addressed your comments raised in a previous round of review and you feel that this manuscript is now acceptable for publication, you may indicate that here to bypass the “Comments to the Author” section, enter your conflict of interest statement in the “Confidential to Editor” section, and submit your "Accept" recommendation.

Reviewer #1: All comments have been addressed

Reviewer #3: All comments have been addressed

2. Is the manuscript technically sound, and do the data support the conclusions?

Reviewer #1: Yes

Reviewer #3: Yes

3. Has the statistical analysis been performed appropriately and rigorously? 

Reviewer #1: Yes

Reviewer #3: Yes

4. Have the authors made all data underlying the findings in their manuscript fully available?

Reviewer #1: Yes

Reviewer #3: Yes

5. Is the manuscript presented in an intelligible fashion and written in standard English?

Reviewer #1: Yes

Reviewer #3: Yes

6. Review Comments to the Author

Reviewer #1: (No Response)

Reviewer #3: This revised version of the manuscript by Liktor-Busa and colleagues has properly addressed the major concerns and comments raised in the first review round. The authors improved the western blot figure 4 and performed new experiments in order to demonstrate BBB integrity in vivo, particularly, using FITC-dextran transport in aCSF and KCl injected rats. I have not further comments and the manuscript should be published as it is.

7. PLOS authors have the option to publish the peer review history of their article (what does this mean?). If published, this will include your full peer review and any attached files.

Reviewer #1: Yes: Botir T Sagdullaev

Reviewer #3: Yes: Juan A. Orellana

---

## [Editor Report · Acceptance letter]

15 May 2020

PONE-D-19-34794R1 

Functional NHE1 Expression is Critical to Blood Brain Barrier Integrity and Sumatriptan Blood to Brain Uptake 

Dear Dr. Largent-Milnes:

I am pleased to inform you that your manuscript has been deemed suitable for publication in PLOS ONE. Congratulations! Your manuscript is now with our production department. 

With kind regards,

on behalf of

Dr. Eliseo A Eugenin 

Academic Editor

PLOS ONE